# Structure defining of ultrapotent neutralizing nanobodies against MERS-CoV with novel epitopes on receptor binding domain

Sen Ma[1]☉, Doudou Zhang[2]☉, Qiwei Wang[1], Linjing Zhu[2], Xilin Wu[2]*, Sheng Ye[1]*, Yaxin Wang📧[1]*

**1** Frontiers Science Center for Synthetic Biology (Ministry of Education), Tianjin Key Laboratory of Function and Application of Biological Macromolecular Structures, School of Life Sciences, Tianjin University, Tianjin, P.R. China, **2** Center for Public Health Research, Medical School, Nanjing University, Nanjing, P.R. China

☉ These authors contributed equally to this work.
* xilinwu@nju.edu.cn (XW); sye@tju.edu.cn (SY); wangyaxin@tju.edu.cn (YW)

**Data Availability Statement:** All data are in the manuscript and supporting information files.

**Funding:** This study was supported by Ministry of Science and Technology to SY under grant number

## Abstract

The Middle East Respiratory Syndrome Coronavirus (MERS-CoV) causes severe and fatal acute respiratory disease in humans. High fatality rates and continued infectiousness remain a pressing concern for global health preparedness. Antibodies targeted at the receptor-binding domain (RBD) are major countermeasures against human viral infection. Here, we report four potent nanobodies against MERS-CoV, which are isolated from alpaca, and especially the potency of Nb14 is highest in the pseudotyped virus assay. Structural studies show that Nb14 framework regions (FRs) are mainly involved in interactions targeting a novel epitope, which is entirely distinct from all previously reported antibodies, and disrupt the protein-carbohydrate interaction between residue W535 of RBD and hDPP4 N229-linked carbohydrate moiety (hDPP4-N229-glycan). Different from Nb14, Nb9 targets the cryptic face of RBD, which is distinctive from the hDPP4 binding site and the Nb14 epitope, and it induces the β5-β6 loop to inflect towards a shallow groove of the RBD and dampens the accommodation of a short helix of hDPP4. The particularly striking epitopes endow the two Nbs administrate synergistically in the pseudotyped MERS-CoV assays. These results not only character unprecedented epitopes for antibody recognition but also provide promising agents for prophylaxis and therapy of MERS-CoV infection.

## Author summary

MERS-CoV is one of the most prevalent zoonotic virus that has spread through 27 countries and infected more than 2,605 people since its first outbreak in Saudi Arabia in 2012. The high fatality rate and its persistent wide spread infectiousness in animal reservoirs have generated tremendous global public health concern. However, no licensed therapeutic agents or vaccines against MERS-CoV are currently available. To address the risk of MERS-CoV and its variants re-emergence, we developed four highly effective Nbs from alpaca and solved two crystal complexes of Nb9 and Nb14 with RBD. Remarkably, the

2020YFA0908500 (https://service.most.gov.cn/), the National Natural Science Foundation of China to SY under grant number 31971127 (http://www.nsfc.gov.cn/), and to YW under grant number 81801998 (http://www.nsfc.gov.cn/). The funders had no role in study design, data collection and analysis, decision to publish, or preparation of the manuscript.

neutralizing activity of Nb14 represents the highest of MERS-CoV antibodies reported to date. The novel epitopes of Nb14 and Nb9 locate outside the RBD and hDPP4 interface, and the unique different epitopes and mechanisms of Nb14 and Nb9 provide a better neutralizing for synergistically against MERS-CoV infection. Our findings provide insights into the cryptic epitopes on RBD and the development of antibodies against MERS-CoV infection.

## Introduction

Middle East respiratory syndrome coronavirus (MERS-CoV) is a zoonotic virus first identified in Saudi Arabia in 2012 [1]. Despite nearly a decade having passed since it caused two pandemics in Saudi Arabia and South Korea [2–4], cases continue to be reported in the Arabian Peninsula (http://www.who.int/emergencies/mers-cov/en/). MERS-CoV, as well as severe acute respiratory syndrome SARS-CoV and SARS-CoV-2, is classified as beta-coronavirus (CoV) [5], a highly pathogenic human CoV that causes acute respiratory distress syndrome (ARDS) [6]. To be note, among the three virulent viruses, the mortality rate of MERS-CoV infection is estimated to be as high as 35%, far exceeding that of SARS-CoV (10%) and SARS-CoV-2 (<3%) [7]. Different from SARS and SARS-CoV-2 [8], dromedary camels are probably an intermediate host for the infection of MERS-CoV in the humans [9,10]. While measures taken during the COVID-19 pandemic (such as quarantine policies and wearing a mask) may reduce human transmission of MERS-CoV, the circulation of MERS-CoV in dromedary camels is unlikely to be impacted. Thus, with the COVID-19 pandemic being managed and standardized across the globe, MERS-CoV, which is transmitted directly from camels to human and from human to human, continues to trigger an epidemic of high fatality rate in the Arabian Peninsula, and the potential pandemic initiated by travelers from this location remains a serious threat to public health [11].

Like SARS-CoV-2 and SARS-CoV, the MERS-CoV Spike glycoprotein (S) plays a critical role in viral infection and the receptor response. The S protein comprises two functional subunits: the N-terminal S1 subunit forms the globular head, while the membrane-embedded C-terminal S2 region forms the stalk region [12–17]. The crucial S protein initially mediates virus entry into target cells through the interaction of receptor binding domain (RBD) on S1 subunit with the human cellular receptor dipeptidyl peptidase 4 (hDPP4) and subsequently through fusion between the viral envelope and the host cell membrane by the S2 subunit [18–20]. Therefore, disrupting the interaction between RBD and hDPP4 with neutralizing antibodies could be a promising therapy for MERS-CoV infection. Up till now, more than 20 antibodies have been developed, the majority of which target the RBD of S protein, causing steric clashes that interfere with the binding of hDPP4 [21–26]. However, there are neither vaccines nor antiviral therapeutics available to prevent or treat MERS-CoV infection thus far [7,27–31].

Nanobodies (Nbs), single-domain fragments of camelid heavy-chain antibodies named VHH [32], are gradually being accepted in the clinic as antiviral agents [33–37]. Due to their special nature of size, stability, specificity, high affinity, low immunogenicity, and bulk production in cost-effective systems, Nbs have been a desirable candidate for next-generation interventions against CoVs [38–44]. Notably, by leveraging the special properties of Nbs at the nanoscale and their long CDR3 regions, they are more feasible to access the cryptic epitope of antigens. Previously, we developed specific Nbs against SARS-CoV-2 RBD [45], and the Nb22 not only against the wild-type strain but also exhibits potent neutralizing on Delta strain [46]. Accordingly, the development of novel epitope signatures, ultra-high neutralization activity,

and broad-spectrum therapeutic Nbs represents an effective approach to tackle the looming crisis posed by the MERS-CoV pandemic.

In this study, we identify four potent Nbs (named Nb9, Nb11, Nb14, and Nb67) capable of specific binding to the recombinant S of MERS-CoV (2012/EMC), through screening a nanobody library from immunizing an alpaca. The $IC_{50}$ against MERS-CoV reaches as low as a single or less nanomolar concentration, representing potency and specificity of the four Nbs described to date. Particularly, Nb14 displays the highest neutralizing potency ($IC_{50}$ value of 0.0014 μg/ml) against MERS-CoV pseudovirus, and Nb9 exhibits a unique epitope that distinguishes it from other Nbs. Crystal structure analysis reveals that Nb14 framework regions (FRs) are mainly involved in interactions with the RBD and describes unprecedented epitopes rarely reported previously [7]. Nb14 appears to exert its neutralizing activity by blocking the RBD-hDPP4-N229-glycan interaction. Differently, Nb9 targets the cryptic and outer face of the RBD, which is distinctive from the hDPP4 binding site and the Nb14 epitope, inducing the RBD conformation change to exert neutralizing activity. The unique striking epitope and mechanism of Nb14 and Nb9 provide better neutralization synergistically against MERS-CoV infection. These results indicate that we have identified two different cryptic epitopes of Nbs that play completely different mechanisms of action. The nanobodies and the epitopes identified should thus provide an invaluable reference against the crisis of potential MERS-CoV transmission and pandemic.

## Results

### Generation and isolation of MERS-CoV S-Directed Nbs

We firstly immunized the alpaca with extracellular domains of MERS-CoV S protein (S1+S2 ECD) to generate the specific anti-sera response (Figs 1 and S1A). Compared to the pre-immunized serum (blank serum), the anti-serum after the fourth immunization exhibited specific serologic activities against MERS-CoV S proteins, with an anti-serum titer of ~$10^6$ dilution (S1B Fig). The fourth immunoserum exhibited potent neutralization activity against the pseudotyped MERS-CoV, with a half-maximal neutralization dilution ($ND_{50}$) of ~9700 (S1C Fig).

We subsequently constructed a phage VHH library (MERS-VHH-lib) based on the cDNA of peripheral blood lymphocytes from the last immunized alpaca (Figs 1 and S2A). The library had a size of $1.37 \times 10^9$, exhibiting a remarkable sequence diversity of 100% with an impressive in-frame rate of 84%, which was rigorously validated through PCR and sequencing. We then analyzed the library binding to the S protein and found an incremental increase in the optical density 450 (OD450) readout, from 0.79 before enrichment to 1.0, 2.6, and 3.4 after the first, second, and third rounds of enrichment, respectively (S2B Fig). To verify the specificity of S reactive phages, single clones were selected from the libraries following the first, second, and third rounds of enrichment for single-phage ELISA. Specifically, the percentage of positive clones was 32.5% and 42.9% for the second and third rounds, respectively (S2C Fig). These results indicated that we successfully immunized alpacas and established a VHH phage library, enabling the isolation of S-positive single clones after three screening rounds.

### Character and determination of Nbs with potent neutralization activity against MERS-CoV

Four Nbs, named Nb9, Nb11, Nb14, and Nb67, exhibited the optimum neutralizing activities among 182 positive clones and were selected for further evaluation (S2D Fig). We initially charactered the Nbs′ affinity with the S by fluorescence-activated cell sorting (FACS), the flow cytometry results demonstrated that the Nb-Fcs (which were fused with a human Fc1 in Nbs′

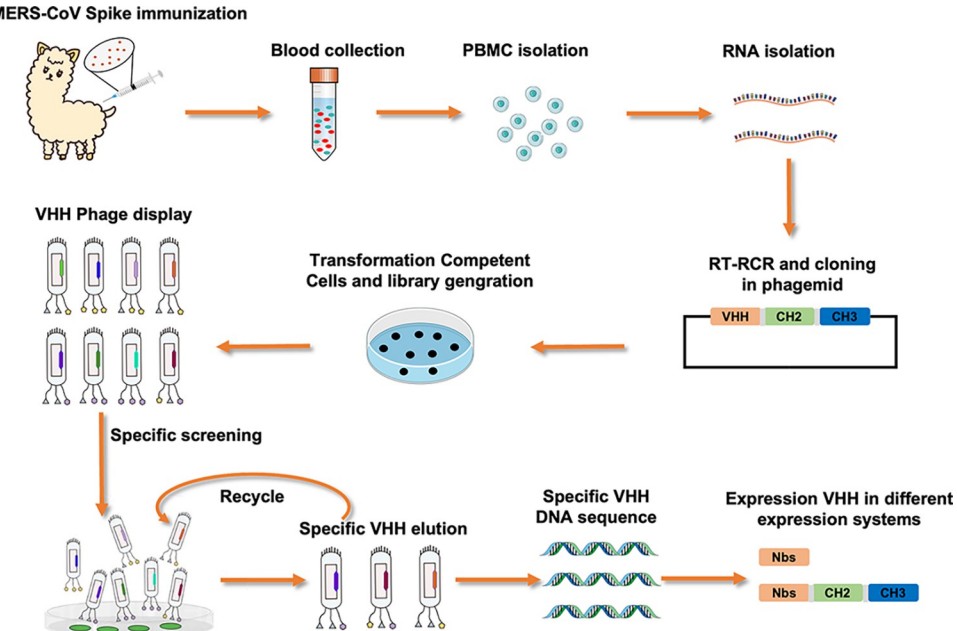

**Fig 1. Schematic diagram establishment of MERS-CoV nanobody screening library.** Immunized the alpaca with extracellular domains of MERS-CoV S protein and collected the blood after the final immunization to isolate PBMCs. Extracted RNA to synthesize cDNA via RT-PCR, and established phage screening library. After four rounds of bio-panning, the specific VHH coding sequences were confirmed from the selected positive clones and further assayed using candidate nanobodies. See also S1 Fig.

C-terminal) had strong binding affinities to 293T cells which overexpressed the S protein (Figs 2A and S3B). To map the recognized region of Nbs, we subsequently determined the affinity to MERS-CoV S1 subunit by Bio-Layer Interferometry (BLI) and tested the binding affinity to RBD [15,22] and N-terminal domain (NTD) [47,48] with enzyme-linked immunosorbent assay (ELISA) and pull-down assay. The Nb-Fcs demonstrated strong binding to MERS-CoV S1, exhibiting a dose-dependent manner (Fig 2B). Additionally, BLI binding analysis revealed the kinetic binding affinity ($K_D$) values of 1 pM for Nb9-Fc, Nb14-Fc, and Nb67-Fc with the S1 subunit, whereas for Nb11-Fc was 0.1 nM, respectively (Figs 2C and S3A). Following, we evaluated the epitope specificity through BLI competition experiments using the S1 subunit as a capture antigen. The results revealed that the pre-bound Nb9-Fc efficiently combined with other Nb-Fcs to the S1 subunit, suggesting that Nb9 recognized a unique epitope distinct from the other three Nbs (S3C Fig). Furthermore, to investigate the specific recognized region of Nb9, ELISA was utilized to detect the specificity of Nbs for NTD and RBD. As anticipated, although Nb9 was detected as a distinctive epitope by BLI, Nb9 and other Nbs exhibited specific binding to the RBD and nearly identical reactivity with the MERS-CoV S1 (Fig 2D). The pull-down assay also proved Nbs bind to the RBD and formed a complex (Fig 2G). These results indicated that RBD was the recognized region of these four Nbs, of which Nb9 demonstrated a distinctive epitope independent of other Nbs.

To assess the neutralizing efficacy of various Nbs, we enveloped pseudotype MERS-CoV bearing the EMC strain S protein by HEK-293T cells and conducted assays on Huh-7 cells and Vero E6 cells. We gradient-diluted the Nbs purified from *E. coli* and subsequently tested the neutralization assays. The neutralizing activity of Nbs was dose-dependent in Huh-7 cells, and with stronger potency of Nb14 (IC$_{50}$, 0.0014 µg/ml), Nb11 (IC$_{50}$, 0.0023 µg/ml), and Nb67 (IC$_{50}$, 0.0025µg/ml), respectively (Fig 2E and 2H). In contrast, Nb9 displayed a slightly weaker

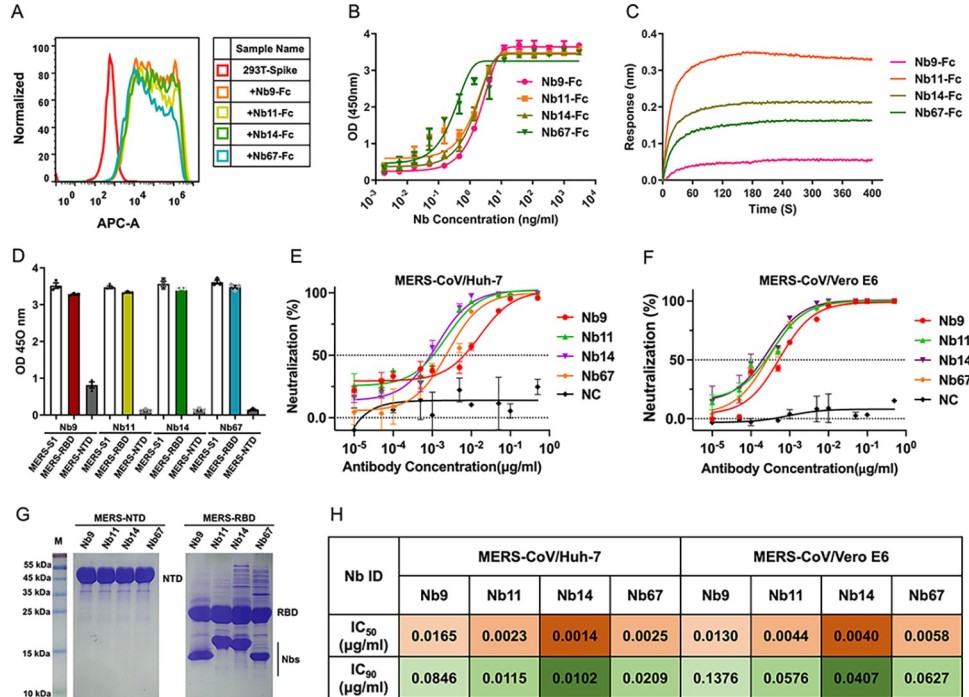

**Fig 2. Determination and Characterization of MERS-CoV S protein-specific Nbs. (A)**. Characterized the affinity of Nbs bound to MERS-CoV S protein by Flow Cytometry. HEK-293T cells, which overexpressed the MERS-CoV S protein, stained by Nb9-Fc, Nb11-Fc, Nb14-Fc, and Nb67-Fc. HEK-293T cells were washed by PBS (Solarbio) and stained by anti-Fc-APC (Abcam). (**B**). The binding of purified Nbs-Fc with the MERS-CoV S1 subunit was identified using ELISA. Data represented as mean ± SD. (**C**). The Kinetic binding curves of the MERS-CoV S1 subunit with Nb9-Fc, Nb11-Fc, Nb14-Fc, and Nb67-Fc were detected by BLI. (**D**). Determination of the MERS-CoV S1 subunit recognition region of Nbs by ELISA. (**E, F**). Nb9, Nb11, Nb14, and Nb67 could effectively neutralize MERS-CoV pseudovirus in vitro. MERS-CoV pseudovirus was incubated with serially diluted Nbs and detected luciferase activity in Huh-7 cells (E) and Vero E6 cells (F), respectively. The neutralization potency of each Nbs was evaluated in a luciferase assay system. Nb22 was taken as a negative isotype control antibody (NC). Data are represented as mean ± SD from three independent experiments. All experiments were repeated at least twice. (G) Determination of the MERS-CoV S1 subunit recognition region of Nbs by Pull-Down. (H) The neutralization potencies of Nbs. The IC50 and IC90 were labeled orange and cyan, respectively. The intensity of the color indicates the strength of the neutralization activity. ELISA, enzyme-linked immunosorbent assay; BLI, Bio-Layer Interferometry; Nbs, nanobodies; MERS-NTD, N-terminal domain of MERS-CoV Spike glycoprotein; MERS-RBD, receptor-binding domain of MERS-CoV Spike glycoprotein; MERS-CoV, Middle East Respiratory Syndrome Coronavirus.

neutralizing activity with an IC50 of 0.0165 μg/ml (Fig 2E and 2H). The neutralizing potency of the nanobodies showed consistent results in Vero E6 cells. The neutralizing activity of Nb14 was the best, with an IC50 of 0.0040 μg/ml, and practically the same with Nb11 and Nb67 with an IC50 of 0.0044 μg/ml and 0.0058 μg/ml, respectively (Fig 2F and 2H). Similarly, the potency of Nb9 was weaker than other Nbs, with an IC50 of 0.0130 μg/ml (Fig 2F and 2H). Although the results of the tests vary, the results all show that Nb14 has the best neutralization effect. In addition, the IC50 values of Nb14 significantly exceed the ultrapotent neutralizing antibodies or other Nbs reported in the pseudotyped virus assay. Thus, for all assays employed, we successfully isolated four Nbs that exhibited robust binding to MERS-CoV RBD and effectively blocked MERS-CoV entry into susceptible cells.

## Structure definition of Nbs bound to MERS-CoV RBD

To dissect the neutralization mechanisms of these Nbs and to better define the epitopes that they recognized, we purified the complexes of Nbs-RBD and attempted to solve the structures

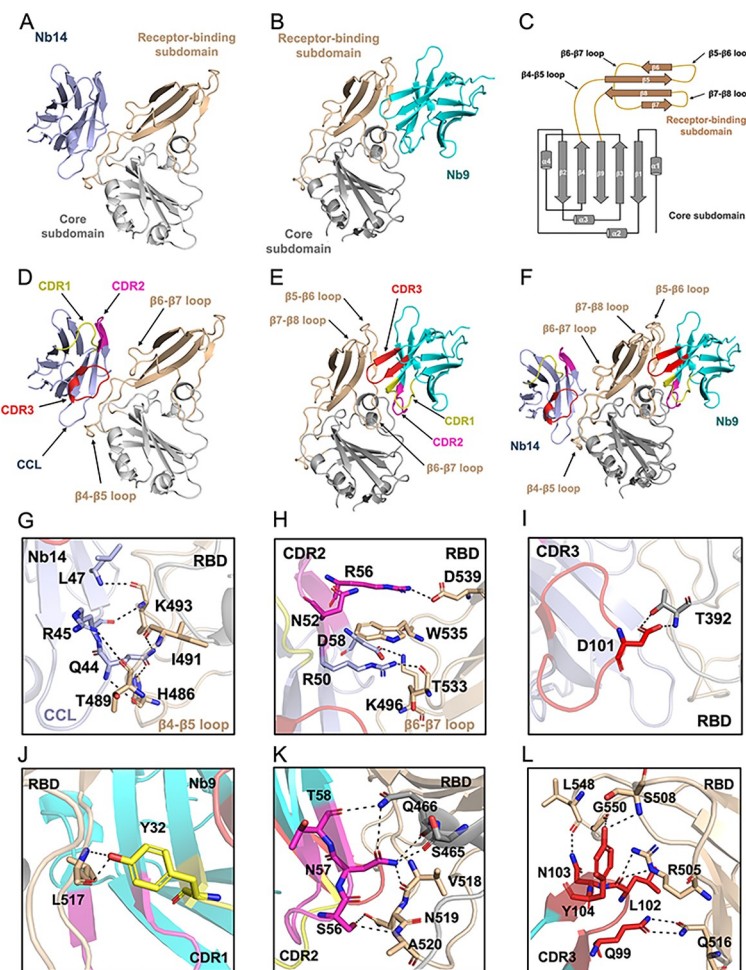

**Fig 3. Crystal structure and binding interface of Nb9 and Nb14 bound to MERS-CoV RBD.** (**A, B**). Crystal structures of Nb14 and Nb9 bound to MERS-CoV RBD, respectively. Receptor-binding subdomain, core subdomain, Nb14, and Nb9 are colored in wheat, gray, slate, and cyan, respectively. (**C**). Schematic illustration of MERS-CoV RBD topology. β strands are drawn as arrows and α helices are drawn as cylinders. (**D, E**). The interaction region of Nbs and RBD. (CDR1: yellow, CDR2: pink, CDR3: red, and CCL: slate). (**F**). The structure superimposition of Nb14-RBD and Nb9-RBD. (**G, H, and I**). Interactions between the residues of RBD and Nb14. The CCL recognized to the β4-β5 loop of RBD (G); The CDR2 of Nb14 was recognized with the β6-β7 loop of RBD (H); CDR3 bound to RBD (I). (**J, K, and L**). Interaction residues between RBD and Nb9. The CDR1 bound to the β6-β7 loop of RBD (J), the CDR2 bound to the β5-β6 loop of RBD (K), and the CDR3 recognized with β5-β6 loop and β7-β8 loop of RBD (L). CCL, Nb14 complementarity cage loop (FRs of Nb14 aa: G42-L47); CDR, complementarity determining region.

utilizing X-ray crystallography (S4 Fig). Although we could not characterize the atomic level of the Nb11-RBD and Nb67-RBD complex with vast screening and optimization, crystal structures of Nb14 and Nb9 bound to the RBD complexes were successfully determined, respectively (Fig 3A and 3B). The structure of Nb14 bound to RBD was resolved at 1.99 Å resolution with a final $R_{work}$ of 0.229 and $R_{free}$ of 0.239 (PDB: 8YSH), whereas the structure of Nb9 bound to the RBD was solved to 2.76 Å resolution with a final $R_{work}$ of 0.233 and $R_{free}$ of 0.266 (PDB: 8YSF). Statistics of diffraction data collection, processing, and structural refinement were listed in S1 Table, and the final models contained one Nb14-RBD complex and two Nb9-RBD complexes per asymmetric unit, respectively.

To our surprise, the epitopes of Nb9 and Nb14 were entirely different despite both binding to the receptor-binding subdomain (RBM) of RBD (Fig 3A and 3B). The previous study

showed that MERS-CoV RBD folds into an RBM, which can bind to the human receptor hDPP4, and a core subdomain (Fig 3C). Upon binding, Nb9 and Nb14 contact with the RBM of RBD (S5B Fig), and all three CDRs of Nb9 bound to RBD with a cryptic and outer epitope with a buried surface area (BSA) approximately 912.2 $Å^2$ (Figs 3E and S5A). Although the four constant FRs are responsible for maintaining the structural integrity of Nbs and barely involved in receptor recognition [49], Nb14 complementarity cage loop (CCL, FRs of Nb14 aa: G42-L47) specifically recognized and interrelated with the RBD. Exquisitely, CDR2 and CDR3 of Nb14 are binding with RBD, and the CCL of Nb14 interrelated with the β4-β5 loop of RBD instead of CDR1 recognized (Figs 3D and S5A). The BSA between the Nb14 and the RBD around approximately 845.6 $Å^2$, smaller than that of Nb9. Further structure superimposition of Nb14-RBD and Nb9-RBD complex showed that the epitopes of Nb14 and Nb9 independently map on the opposite sides of the RBD and recognized non-intersecting residues (Figs 3F, S5B and S5C).

### Distinct epitope recognized features of Nb14-/Nb9- RBD complex

To decipher the epitope-recognition features, we analyzed the binding interface of the Nb14-/Nb9- RBD complex. Distinguishing from CDRs of the Nb9 serve as the primary regions for RBD recognition, an intriguing discovery emerged where the CCL of Nb14 formed a classic β-sheet parallel structure with the β4-β5 loop of the RBD contributing to the core interaction interface (Figs 3G and S5D). In the framework of the β-sheet parallel structure, the Nb14 residues Q44, R45, and L47 intricately interacted with specific counterparts in the RBD, including T489, I491, and K493, establishing a robust and comprehensive interaction interface (Fig 3G and S2 Table). The residues Q44 of the CCL in Nb14 formed two distinct hydrogen bonds, one with RBD residues H486 and the other with I491. Similarly, R45 in Nb14 established two crucial hydrogen bonds with RBD residue K493 and T489, respectively. L47 in Nb14 was also specifically engaged in a hydrogen interaction with RBD residue K493 (Fig 3G). Furthermore, the residues G42, K43, and E46 in CCL of Nb14 interacted with RBD in hydrophobic and significantly contributed to the structural stability of the complex (S5E Fig).

Apart from the CCL of Nb14, the residues of R50, R56, D58 in CDR2, and D101 in CDR3 were also involved in the hydrogen bonds interaction with RBD (Fig 3H and 3I and S2 Table). The R50 and D58 in proximity to CDR2 of the Nb14 especially engaged in interactions with RBD residues T533 and K496 respectively, and collectively formed an interaction network within CDR2 (Fig 3H). Within the CDR2, the residue R56 of Nb14 formed hydrogen bonds and salt bridges with the RBD residue D539, and N52 of Nb14 engaged in hydrophobic interactions with RBD residues W535. CDR3, however, only provided a diminutive contribution via two hydrogen bonds, forming by D101 of Nb14 and T392 of RBD (Fig 3I).

Diverging from the Nb14 binding mode, the interaction between Nb9 and RBD adhered to a conventional antibody paradigm, emphasizing the pivotal role of CDRs in Nb9 antigen recognition. The binding interface between RBD and Nb9 was composed of 12 residues derived from RBD and 10 residues contributed by Nb9. The 8 residues from all three CDRs of Nb9 formed extensive interplay with 10 residues derived from β5-β6, β6-β7, and β7-β8 loops of RBD (S5A–S5C Fig), and the Nb9 FRs residues Y37 and R59 also formed two hydrogen bonds with RBD residues E513 and Q466 respectively (S5F Fig and S3 Table). The Y32 of the CDR1 in Nb9 formed two hydrogen bonds with RBD residues L517 (Fig 3J and S3 Table). The S56, N57, and T58 of the CDR2 in Nb9 interacted with RBD residues A520, Q519, V518, S465, and Q466, and formed a tight and wide network of interactions (Fig 3K). Particularly, the N57 of Nb9 formed an amazing four hydrogen bonds with RBD residues S465, Q466, and V518 respectively, and provided a main network within CDR2 (Fig 3K and S3 Table). The Q99,

L102, N103, and Y104 of the CDR3 in Nb9 bound to RBD residues Q516, R505, L548, G550, and S508, and established an intricate interface of interactions (Fig 3L and S3 Table). In conclusion, The CDR2 and CDR3 regions exhibit pronounced interaction with the RBD, thereby crucially maintaining the structural integrity of the recognition interface.

### Confirmation of the unique neutralizing epitope

The aforementioned crystal structures revealed a significant finding that the Nb14 and Nb9 located on the opposing interface of RBM and exhibited an apparent absence of overlapped residues on RBD-recognition motifs (Figs 4A and S5C). To further confirm the key residues of unique epitopes in binding and neutralization, we performed a mutagenesis study by introducing single mutations to S glycoproteins and then generated a panel of pseudotyped MERS-CoV with the S mutants. The ELISA assays revealed a significant reduction in the neutralization efficacy of both Nb14 and Nb9 against strains carrying specific mutations. There was an approximately 10-fold decrease in $IC_{50}$ values for RBD residues H486A, T489A, I491A, and K493A mutant strains which Nb14 recognized at the CCL recognition interface (Fig 4B and 4D and S4 Table). Especially, the neutralization effect of Nb14 against K493A, L495A, and K496A mutant strains was a 170-fold decrease (Figs 4D and S6A and S4 Table), which means these

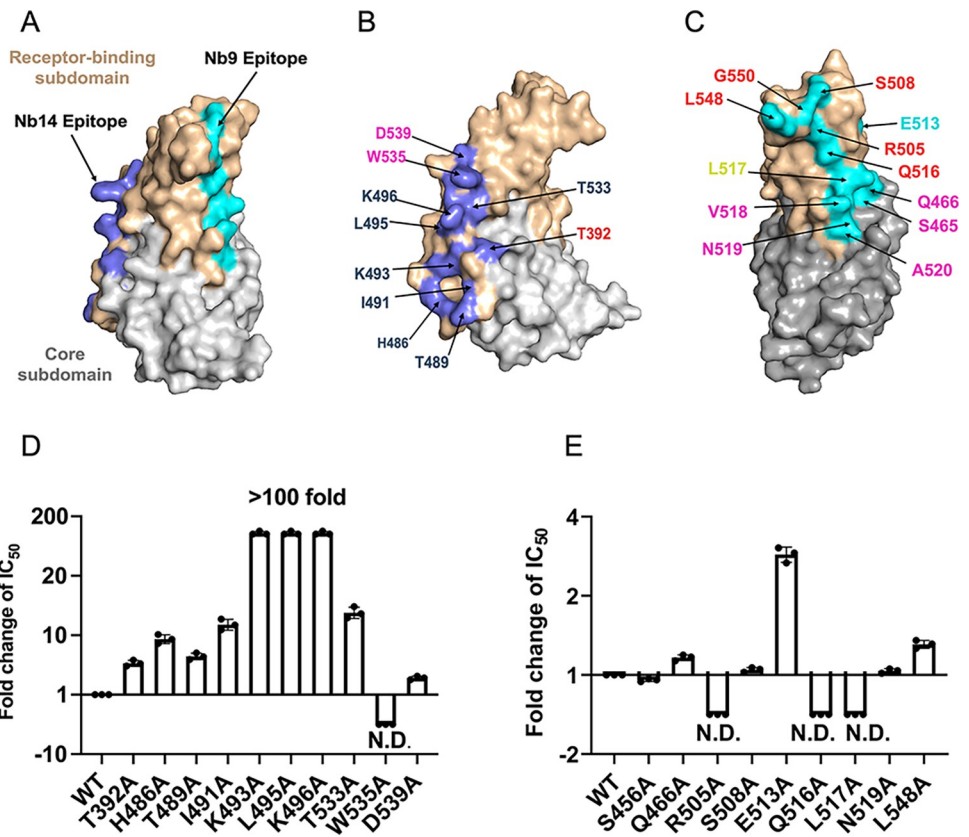

**Fig 4. Confirmation of the neutralizing epitopes.** (**A**). Diverse epitopes of Nb14 and Nb9. Nb14 epitope is slate and the Nb9 epitope is cyan. (**B, C**). Epitopes on the RBD recognized by Nb14 and Nb9, respectively. The pseudovirus of MERS-CoV variants identified from the Nb14 and Nb9 epitopes were tested to evaluate the neutralization potency conferred by Nb14 (**D**) and Nb9 (**E**), respectively. The names of MERS-CoV variants with an amino acid mutation based on the wild type (WT) of MERS-CoV are indicated. The y-axis shows the ratio of $IC_{50}$ of indicated variant/$IC_{50}$ of WT conferred by Nb, and N.D. means not detected. Data are represented as mean ± SD from three independent experiments.

residues were critical for Nb14 recognized RBD. Furthermore, we also performed a mutagenesis study by introducing single mutations to all recognized residues from Nb14 bound to RBD, and we found all the Nbs mutations had variant unequal effects on the binding by reducing the affinity in the range of 1000-fold, especially R50A, R56A, D58A, and T60A of Nb14 dramatically decreased (S5C and S7A Figs and S5 Table), which consistent with the results reflected in neutralization of Nb14 against pseudotyped MERS-CoV mutants.

In parallel, Nb9 also displayed notable attenuation of $IC_{50}$ values when encountering mutations within its recognition epitope. The results revealed a 3-fold decrease in neutralization potency against the E513A mutant. Moreover, the neutralization activity of Nb9 was not detectable against the R505A, Q516A, and L517A mutants (Fig 4C and 4E). Again, we tested the binding affinity between mutants of Nb9 and RBD, and the BLI results showed that most targeted mutations in Nb9 also aggravated affinity by approximately ~4 orders of magnitude, of which, the $K_D$ of Q99A was abrogated to completely unbound (S5B and S5D Fig and S5 Table). Collectively, these results mutually reinforce and conclusively affirm the distinct epitope of Nb14 and Nb9 delineated in our study.

## Nb14 and Nb9 maintain cross-neutralizing activity to pseudotyped MERS-CoV with natural mutations

The relentless tides of MERS-CoV evolution have carved out a multitude of ways to exploit mutations in the RBD protein to escape the neutralization of antibodies. Sequencing of multiple MERS-CoV isolates revealed that the RBD is evolving through substitutions, and the change in RBD for CoVs was critical which influenced the host switching and virus pandemic [50]. To assess the cross-neutralizing activities of Nb14 and Nb9 against divergent MERS-CoV mutations, we identified the mutation residues from alignment to the EMC, and focused on the mutations of RBD, including 8 changing residues: L411F, T424I, L506F, D509G, I529T, V530L, E536K, D537E, which were isolated from humans in Saudi Arabia, England, and South Korea. Particularly, these natural changes, named L506, D509, and E536, could potentiate the MERS-CoV to escape the neutralization of antibodies targeting the RBD [11,24]. Interestingly, we mapped the natural mutants in RBD, and the results showed that these substitutions were located either outside or proximate to Nbs epitopes, suggesting that the epitopes targeted by Nb14 and Nb9 were extremely conserved and not susceptible to mutations (Fig 5A–5C). Pseudoviruses bearing these mutations were constructed and tested against a serial dilution of Nb14 and Nb9. Among the viable mutants tested, Nb14 was mostly affected by single D537E and I529T substitutions (Fig 5B), which were located near the Nb14 epitope, and the $IC_{50}$ against D537E and I529T substitutions was 4.31- and 3.08-fold compared to WT respectively (Fig 5D and 5F). Nb9 maintained potency to D537E and I529T mutations, however, was reduced the neutralization by the single mutation L506F and D509G, which were located at the β5-β6 loop on RBD. Compared to WT, the $IC_{50}$ of L506F and D509G mutation was dropped 4-fold, respectively (Fig 5E and 5F). In a word, the Nb14 and Nb9 had broad-spectrum cross-neutralizing activity against naturally occurring mutations in MERS-CoV strains, although the neutralization decreased in some mutations.

## Diverse mechanism of action among the two nanobodies

With the detailed structure and functional information in hand, we spontaneously wondered how Nbs block the viral infection as the epitopes targeted by Nb14 and Nb9 were located outside the hDPP4 binding surface on the RBD (Fig 6A). To test whether Nbs directly inhibit the binding of RBD to hDPP4, we evaluated the capacity of Nbs to block RBD bound to Huh-7 cells, which overexpressed hDPP4 on the surface of cell membranes, via flow cytometry. The

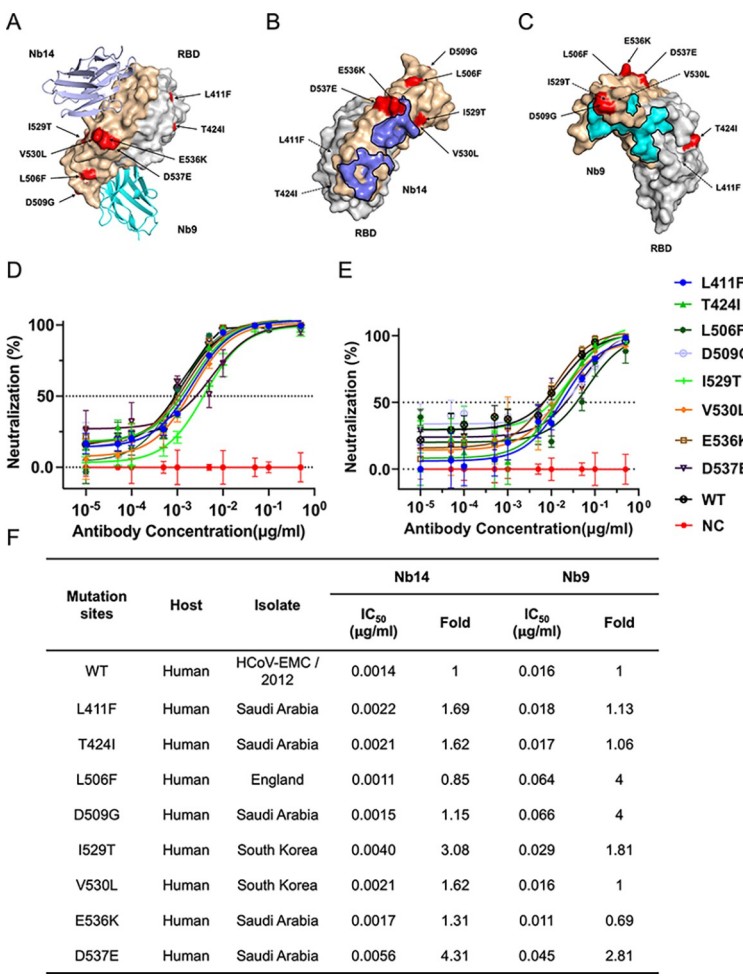

**Fig 5. Neutralizing effects of Nb9 and Nb14 on different MERS-CoV strains. (A)**. The binding modes of Nb14 (slate) and Nb9 (cyan) are binding modes to prototype MERS-CoV RBD with the natural mutation sites highlighted in red. (**B, C**). The epitopes of Nb14 (slate) and Nb9 (cyan) on the surface of MERS-CoV RBD, relative to relevant natural mutation sites highlighted in red. (**D, E**). Neutralizing potencies of Nb14 and Nb9 against pseudoviruses bearing the natural mutations, respectively. Data are presented as the means ± SD from three independent experiments. (**F**). Summary of Nb14 and Nb9 neutralizing activities. $IC_{50}$ neutralization titers for natural mutant variants are presented relative to wild type (WT) of MERS-CoV pseudoviruses.

FACS analysis of cell-surface staining showed that Nb14 and Nb9 inhibited the staining of Huh-7 by the MERS-CoV-RBD, and the staining of RBD was reduced by 93% and 78.3%, respectively (Figs 6B and S10). These results suggested that the Nb14 and Nb9 blocked the RBD bound to hDPP4 and verified the neutralizing effect of Nb9 was lower than that of Nb14. We also superimposed the structures of reported neutralizing antibodies targeted MERS-RBD and divided them into three types: Class I, Class II, and Class III (Figs 6A, S9A and S9B). Class I and Class II antibodies overlapped with the binding interface of the hDPP4-RBD complex and steric clashed with hDPP4 (S9C and S9D Fig). Contrarily, Class III antibodies were situated at the periphery of the hDPP4-RBD binding interface, inducing a conformational alteration in the RBD to hinder hDPP4 binding (S9E Fig). From the structural superimposition, it was interesting to find that the epitope of Nb14 was largely independent of all three class antibodies (S9C–S9E Fig).

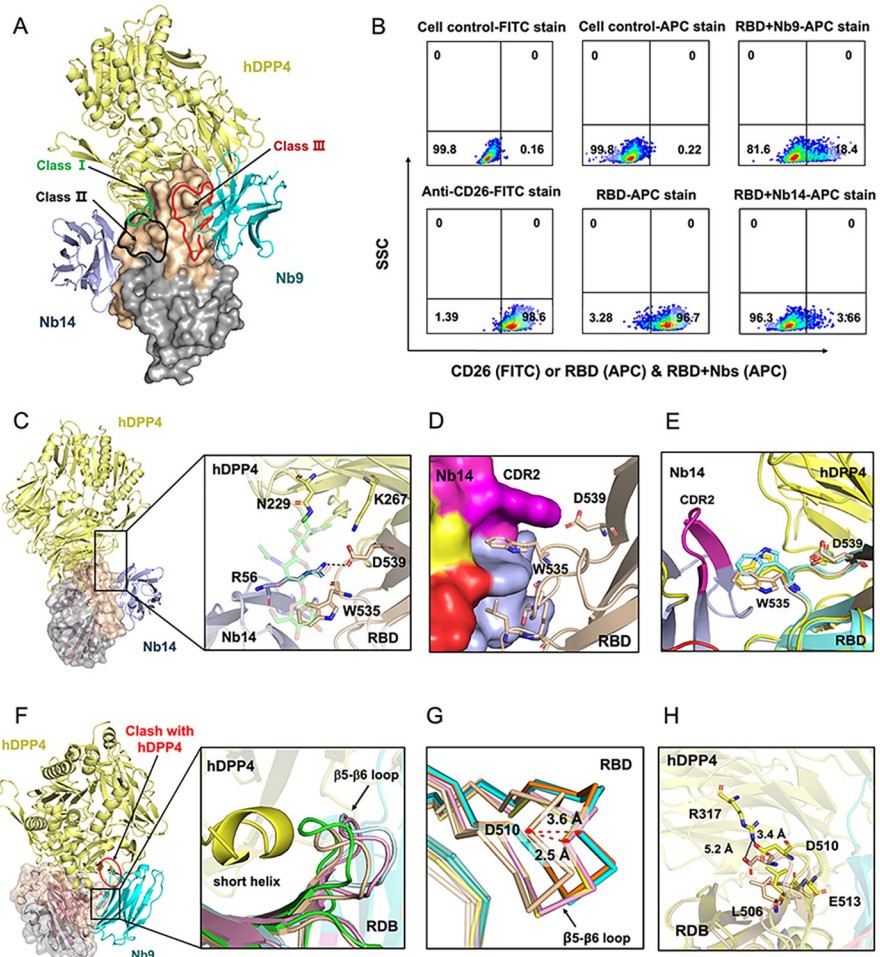

**Fig 6. Mechanism of Nb14 and Nb9 with different neutralizing epitopes.** (**A**). Division of diverse class epitope regions of MERS-CoV RBD, and model structures of hDPP4 (PDB, 4L72; yellow), Nb14 (slate), and Nb9 (cyan) bound to MERS-CoV RBD. (The epitope of class 1: green; class 2: red; and class 3: black; see also in S9 Fig). (**B**). Competitive bindings of monomeric Nb14 and Nb9 with RBD to hDPP4 were measured by Huh-7 cell surface staining. Huh-7 cells were incubated with RBD and Nb14 or Nb9, followed by staining with anti-His-APC (RBD) and analyzed by FACS. The cell staining was repeated, and one representative result was shown. (**C, D, and E**). Neutralization mechanism of Nb14. Structural superimposition demonstrates that the epitope of Nb14 is distinct from the hDPP4 binding site, but clashes with the N229 glycan of hDPP4 recognized to the RBD (**C**). The N229 glycan shows the sticks with green (PDB, 4L72). (**D**). The CDR2 of Nb14 connected to RBD with a tight and wide range interface and formed a cavity that adapted to the W535 of RBD. (**E**). The conformation changes of W535 and D539 in different state RBD. MERS-CoV RBD (cyan), Nb14-RBD (wheat), and hDPP4-RBD (yellow). (**F, G, and H**). Structure definition of the neutralizing mechanism of Nb9. Comparison structure superimposition shows conformational change of the RBD β5-β6 loop in the Nb9-Bound State (F). Nb9 clashes with hDPP4 (red labeled), and the β5-β6 loop exhibits an inward curved conformation to prevent the approach of the hDPP4 short helix when bound to RBD. MERS-CoV RBD (cyan), Nb9-RBD (wheat), MERS-4V2 scFv (green), and hDPP4-RBD (RBD, Magenta; hDPP4, yellow; PDB, 4L72). (**G**) Zoom-in view of the aligned RBD β5-β6 loops in unbound (4KQZ: cyan; 4L3N: orange) and hDPP4-bound (4L72: yellow) with the Nb9 bound (wheat) state. (**H**). The residues of L506, D510, and E513 from the RBD β5-β6 loop exhibit conformational changes when bound to Nb9, interrupting the interaction between hDPP4 and RBD. In hDPP4-RBD (PBD, 4L72), the residues show with yellow; and in Nb9-RBD, the residues show with wheat.

How did Nb14 inhibit MERS-CoV infection, considering its non-overlapping structure with hDPP4 as shown in Fig 6C? Previous research has highlighted the significance of residues W535 and D539 within the RBD for hDPP4 binding and viral entry. Specifically, W535 establishes a robust van der Waals stacking interaction with the hDPP4-N229-glycan, while D539

forms a hydrogen bond with K267 in hDPP4 [23] (Fig 6C). Our structural analysis revealed that the residue R56 in CDR2 of Nb14 formed a critical hydrogen bond with D539 of the RBD, thereby impeding the interaction between D539 and K267 residues of hDPP4 (Fig 6C). More importantly, the CDR2 of Nb14 connected to RBD with a tight and wide range interface and formed a cavity that adapted to the W535 of RBD, and then interrupted RBD recognized to the hDPP4-N229-glycan (Fig 6D). In addition, we found that W535 in the Nb14-RBD structure exhibited a significant rotation and location movement, compared with WT-RBD and hDPP4-RBD (Fig 6E), which suggested that W535 allosteric was critical to the neutralizing mechanism of Nb14. In brief, these results indicated that Nb14 recognized a novel epitope and inhibited MERS-CoV invasion by blocking protein-carbohydrate interactions and competing with the RBD key residues D539.

As the class III antibody, the structure superimposition demonstrated that the Nb9-RBD bound resulted in a conformational change of the RBD β5-β6 loop (L506 to E513), which was critical for accommodating a short helix of hDPP4 [15]. Moreover, in the Nb9-RBD complex structure, the Nb9 bound to RBD possessed a mortise and tenon structure, and this tight binding model allowed Nb9 to reserve enough structure to steric clash with hDPP4 (Fig 6F), which was different from the pre-reported antibodies [25]. We compared the conformational changes in the β5-β6 loop of the RBD between our structure and both bound and unbound hDPP4 structures. The distance change occurred at D510, whose Cα atom moved more than 2.5 Å into the groove (Fig 6G). Moreover, the residues L506, D510, and E513 from this loop exhibited conformational changes, and these residues were involved in forming the core hydrophobic and peripheral hydrophilic interactions with hDPP4 (Fig 6H). We superimposed and compared the structures of Nb9-RBD and 4V2 scFv-RBD, and the results showed that the β5-β6 loop exhibited an inward curved conformation to prevent the approach of the hDPP4 short helix. Particularly, the Nb9-RBD β5-β6 loop exhibited a more inward curved conformation (Fig 6F), which could be related to the deeper binding of Nb9 and RBD. In summary, Nb9-RBD bound not only induced the β5-β6 loop conformation changes to impede the approach of the short helix of hDPP4 but also gave rise to a steric clash with hDPP4, leading to an indirect disruption of the interaction between RBD and hDPP4 to prevent the MERS-CoV infection.

## Structural basis for synergistic neutralization of Nb14 and Nb9

The cryo-electron microscopy (cryo-EM) structures of the MERS-CoV S trimers, as well as SARS and SARS-CoV-2, have demonstrated that the RBD could go through a hinge-like motion to transition between "up" and "down" conformations [12,18]. Binding to the hDPP4 host receptor and activating the S trimers is necessary for virus infection, whereas at least one RBD in the "up" position is required. To explore which state of Nb14 and Nb9 can bind and access the epitope, we superimposed Nb14-/Nb9-RBD bound structure to the MERS-CoV S trimers in different conformational states. The results showed that Nb14 could readily bind to RBD regardless of the state of the S trimers (Fig 7). Nevertheless, steric clashes arose between Nb9 and NTD of neighboring monomer when all RBDs were in the "down" position (Fig 7D). It suggested that the epitope of Nb14 was accessible without distinction of RBD conformation. On the contrary, the epitope of Nb9 was sheltered and inaccessible in the inactivated state. This observation also deciphered why the effective neutralization of Nb14 was superior to Nb9.

We similarly compared the binding of Class I, Class II, and Class III antibodies to the closed state of MERS-CoV S trimers by structure superimpose. Our findings indicate that while most antibodies exhibited limited effectiveness in binding to the closed-state MERS-CoV S trimers,

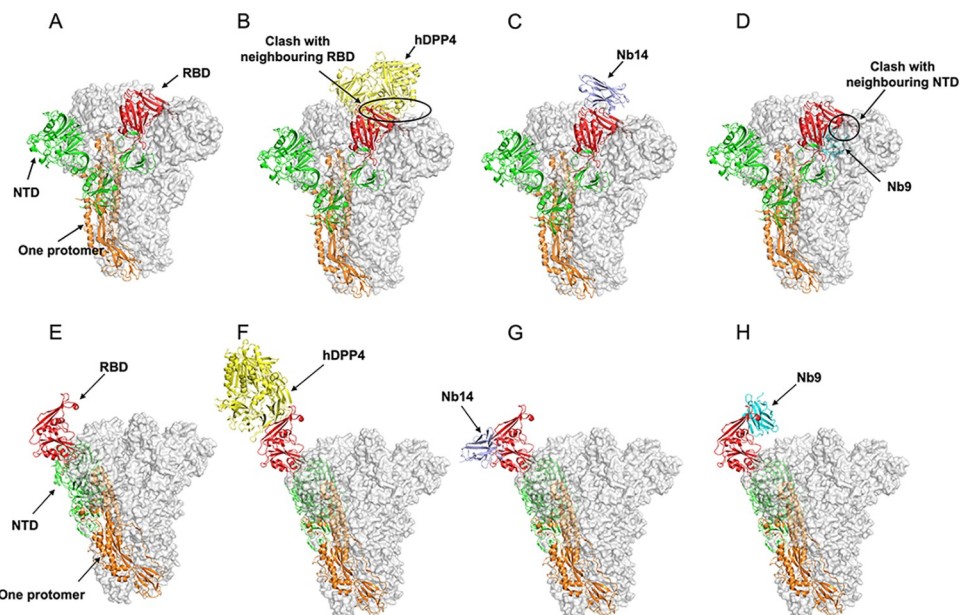

**Fig 7. Structural superimpositions of the hDPP4-RBD, Nb14-RBD, and Nb9-RBD crystal structures onto the MERS-CoV S trimers glycoprotein in Receptor-Binding inactivated and activated states.** (**A**). MERS-CoV S trimers in an inactivated state with all RBD in the "down" positions (PDB: 5w9j). One S protomer is shown as a cartoon (RBD in red, NTD in green, and S2 subunit in orange). (**B, C, and D**). Structure superimpositions of the hDPP4-RBD (B), Nb14-RBD (C), and Nb9-RBD (D), onto inactivated MERS-CoV S trimers, respectively. RBD bind to Nb14 in the inactivated state without a steric clash, but not hDPP4, and Nb9. (**E**). MERS-CoV S trimers in activated state with one RBD in the "up" positions (PDB: 5w9h). (**F, G, and H**). Structure superimpositions of the hDPP4-RBD (F), Nb14-RBD (G), and Nb9-RBD (H), onto activated MERS-CoV S trimers, respectively.

Nb14 demonstrated remarkable capability in engaging with it (S9F–S9H Fig). This robust binding behavior likely underlies the superior neutralizing efficacy of Nb14 compared to other antibodies. Furthermore, we discovered that Nb14 and Nb9 could bind to RBD combination due to the different epitopes between Nb14 and Nb9, suggesting that the neutralization of Nb14 and Nb9 could be synergistic on the activity state of MERS-CoV S trimers. Taken together, these results indicated that Nb14 bound to RBD irrespective of the S trimers conformational states. Nb9, however, bound to RBD in the open state when the MERS-CoV particle transformed to partially activated.

## Synergistic neutralization effects of Nb14 and Nb9

All these results reminded us that the distinct epitopes between Nb14 and Nb9 could be synergistic to neutralize MERS-CoV. To test the synergistic effect of Nb14 and Nb9, we first characterized the RBD whether could simultaneously combine with the Nb14 and Nb9 *in vitro*. The results of size exclusion chromatography (SEC) and SDS-PAGE showed that the RBD could combine with Nb14 and Nb9 simultaneously to form a heterotrimer complex (Fig 8A). As Fig 6B indicated above, the FACS showed that the ability of the Nb9 was weaker than Nb14 to inhibit the staining of Huh-7 by the RBD. To verify whether Nb14 could assist Nb9 to inhibit the binding of Huh-7 by the RBD, we firstly incubated the Nb9 and RBD, then added the Nb14 for further incubation, and finally incubated the mixture with Huh-7 cells, respectively. The FACS showed that the inhibition rates of Nb9 and Nb9+Nb14, which blocked the staining of Huh-7 cells by the RBD, were 80% and 93%, respectively (Fig 8B). When Nb9 and Nb14 fused with human Fc, they were also represented synergistic neutralization effect (S11 Fig).

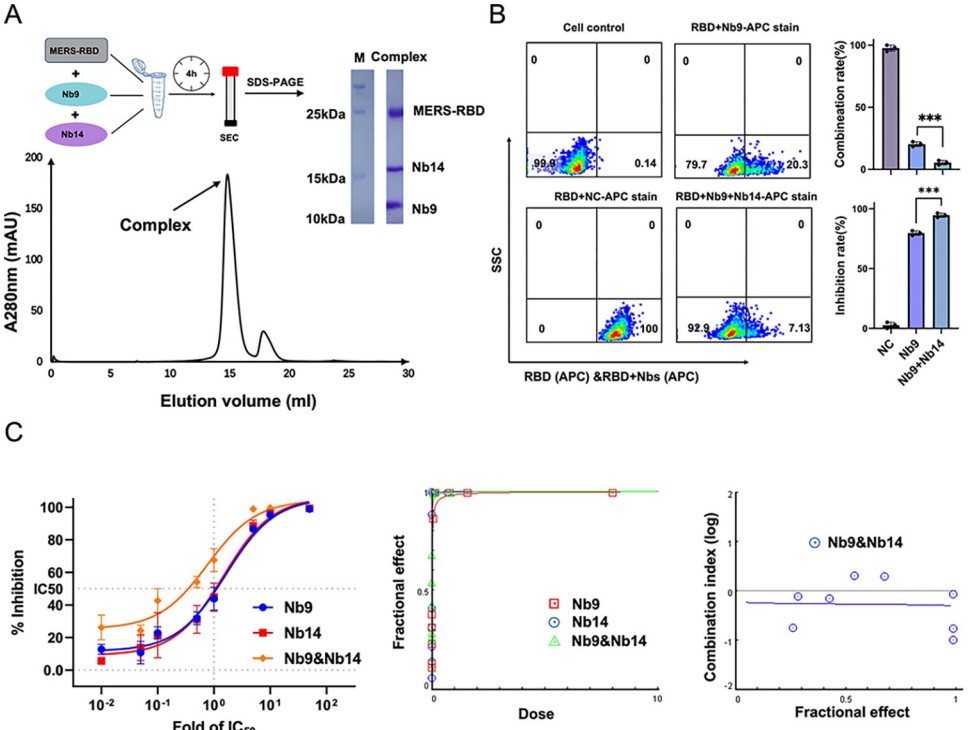

**Fig 8. Synergistic neutralization effects of nanobodies Nb9 and Nb14.** (**A**). The assembly of ternary complexes by Nb9, Nb14, and RBD is purified by size exclusion chromatography (SEC). (**B**). FACS analysis of Nb14 cooperated with Nb9 to inhibit RBD bound to Huh-7 cells. Huh-7 cells were incubated with RBD and then incubated with Nb9 or Nb9 +Nb14, followed by staining with anti-His-APC (RBD) and analyzed by FACS. The combination rate and inhibition rate of RBD to Huh-7 cells significant statistical differences were analyzed by GraphPad Prism 9. Data are represented as mean ± SD from three independent experiments. (**C**). Neutralizing effects of Nb14 combined with Nb9 against pseudotyped MERS-CoV. Nbs were diluted 3-fold and tested alone or in combination to calculate percent neutralization and then combined Nbs at constant ratios. The constant ratios of the combined Nbs were their $IC_{50s}$, and the x-axis means a dose of 1 was at the $IC_{50}$ concentration. Fractional effect (FA) plots were generated by the CompuSyn program and showed a combination of dosage versus effect. Median effect plot of calculated CI values (logarithmic) versus FA values, in which a log CI of <0 is synergism and a log CI of >0 is antagonism. The data shown are average values from three independent experiments.

These results fully illustrated that Nb14 could cooperate with Nb9 due to differences in epitopes.

Subsequently, we tested the synergistic effect of Nb14 and Nb9 at the MERS-CoV pseudo-viruses. As shown in Fig 8C, the neutralization activity for Nb14 and Nb9 combination demonstrated a 2-fold reduction of half maximal inhibitory concentration ($IC_{50}$) compared with single Nb usage. Furthermore, the combination index (CI) values of Nb14 and Nb9 combined at fractional values of effective dose 50%, 75%, 90%, and 95% ($ED_{50}$, $ED_{75}$, $ED_{90}$, and $ED_{95}$) were 0.55, 0.53, 0.52, and 0.51, respectively. As a CI value of 1 indicates an additive effect, CI<1 indicates synergism, and CI>1 indicates antagonism, the combination of Nb14 and Nb9 worked synergistically. Based on our results, it unambiguously proved that Nb14 and Nb9 had a clear synergistic effect when worked together, each of their unique properties combined to create a cumulative effect in the prevention of MERS-CoV infection.

## Discussion

MERS-CoV, which has a high mortality rate (35%) and with dromedary camels serving as its intermediate hosts, still seriously endangers human life and health. At present, the harm of

coronavirus to human health is becoming more and more serious, MERS-CoV, which persists and epidemics in the Arabian Peninsula, must be taken seriously. However, there are neither effective medicines nor vaccines for clinical applications to treat or prevent MERS-CoV infection. Therefore, strategically prioritizing the development of effective countermeasures is imperative to prevent the continued evolution of MERS-CoV and its potential for enhanced human-to-human transmission in the future. Preventative and therapeutic antibodies exhibit the tremendous potential to be effective antiviral therapy. Due to their small size and high neutralization activity, nanobodies are gradually gaining clinical acceptance as antiviral therapeutic agents. To address the high fatality rate and potential pandemic of MERS-CoV, we focus on the development of novel targeted nanobodies.

In this study, four potent neutralizing Nbs were isolated from a phage display platform derived from an alpaca immunized with the MERS-CoV S. We find that these Nbs specifically target the RBD and exhibit remarkable neutralization activities against MERS-CoV. To be noted, Nb14 demonstrates particularly potent neutralizing activity with an $IC_{50}$ value as low as 0.0014 μg/ml against MERS-CoV. Despite variations in neutralization assays among different laboratories, the impressive neutralization level observed *in vitro* suggests that Nb14 is among the broadest and most potent antibodies described to date [7]. Moreover, Nb14 also has strong potency against different mutation strains, and the $IC_{50}$ remains relatively stable across the other mutation viruses tested.

To illuminate the inhibition mechanism, we successfully report the crystal structures of Nb14/Nb9 bound to RBD, and analysis reveals that Nb14 and Nb9 conserve radically different epitopes and neutralization mechanisms. Following, we classify the crystal structures of previously reported antibodies targeting RBD by analogies and analyze the degree of overlap between our Nbs epitopes and the previously reported antibodies. Surprisingly, we find the epitopes of Nb14 and Nb9 between the cryptic and outer faces of RBD, distinctive from the hDPP4 binding location. Not only that, Nb14 recognizes a unique epitope different from all previously reported RBD-targeting antibodies. Although the epitopes of Nb14 and Nb9 are out of the recognition location of hDPP4, the FACS results show that Nb14 and Nb9 disrupt the binding of hDPP4 and MERS-CoV RBD. This highlights the advantage of nanoscale, which enables Nbs to penetrate deeper into the spike and access concealed epitopes that are less reach for conventional antibodies. Although Nb9 and MERS-4V2 scFv share the same epitope motif, as accredited to the nanoscale, the binding model of Nb9 and RBD are more like a mortise and tenon structure, which contributes a more tightly binding interface than the anchoring structure of MERS-4V2 scFv. In the context of this specific structure, we find that the Nb9 clashes with hDPP4 in structure superimposition, which is not found in the structure of the MERS-4V2 scFv-RBD complex.

Reciprocally, framework regions recognized to RBD cause Nb14 to demonstrate a notably distinctive binding modality different from Nb9 and conventional antibodies. Structural features suggest that the CCL of Nb14 plays a significant role in major interaction interfaces, revealing previously unreported epitope features. Sequence analysis further manifests that the RBD residues W535 and D539 overlap between the epitope of Nb14 and the hDPP4 binding location, and these residues, which have been reported to be critical for MERS-CoV invasion in previous studies [23]. We are also pleasantly surprised to discover that the special binding mode of Nb14-RBD allows the interface of CDR2 of Nb14 and RBD to form a cavity to accommodate the residue W535 of RBD, and then directly blocks the recognition between W535 and hDPP4-N229-glycan. Interestingly, the W535 residue in S protein is highly conserved in almost all of the natural MERS-CoV isolates, suggesting that this region is a hot spot for immune recognition and Nb14 possesses the potential to develop into a broad-spectrum anti-MERS-CoV prophylactic and therapeutic agents.

Attribute to the nanoscale size of the Nbs, we discover the important reason that blocks the invasion of MERS-CoV by disrupting hDPP4-N229-glycan recognition. In Lineage C beta-CoVs, HKU-4 can recognize hDPP4 as a cellular receptor for adaptation to infect humans, and the complex structure of HKU4-RBD bound to hDPP4 also illustrates the significant recognition between HKU-4-RBD and hDPP4-N229-glycan [51,52]. These studies suggest that the recognition of hDPP4-N229-glycan plays a crucial role in the invasion of Lineage C betaCoVs and serves as a conserved site of virus identification. Consequently, the neutralizing mechanism of Nb14 holds a significant guidance for the development of broad-spectrum small molecule inhibitors and peptide drugs against Lineage C betaCoVs.

In addition, the combination of antibodies with different epitopes is a very effective treatment. The distinct epitopes demonstrate the feasibility of combining Nb14 and Nb9 to elicit a stronger and broader protective response against MERS-CoV. We analyze the Nb14 and Nb9 bound to MERS-CoV S protein in different conformations by structure superimpose, and the results indicate that Nb14 binds to the epitope that is readily accessible regardless of "up" or "down" conformations of RBD. By contrast, the Nb9 recognizes the hidden epitope only when RBD is in the "up" position. *In vitro* assay shows Nb14 can cooperate with Nb9, and in the pseudovirus test, Nb14 and Nb9 demonstrates a good synergy in inhibiting MERS-CoV infection. These cumulative results also indicate that our Nbs will be useful additions to other formulations of anti-MERS-CoV antibodies, which target non-cross-resistant epitopes and thus decrease the possibility of viral escape. Whereas, due to virus management limitations, we could not further validate the antiviral activity of these two Nbs and combination immunotherapy on live MERS-CoV.

In summary, we have successfully identified and characterized four Nbs displaying potent neutralizing effects against MERS-CoV. Additionally, our research has involved the elucidation of crystal structures for two distinct Nb-RBD complexes with unique epitope characteristics. The discovery of these Nbs holds great promise for clinical therapeutics, offering innovative solutions in response to the persistent and evolving challenges posed by coronaviruses. As the threat of MERS-CoV infection remains a significant concern, it is imperative to advance the development of potential therapeutic interventions. Our study provides a comprehensive and detailed understanding of the mechanisms underlying the actions of these two nanobodies. Furthermore, we have demonstrated a remarkable synergistic effect between them, which not only contributes to our knowledge of MERS-CoV neutralization but also establishes a robust platform for further advancement in antibody-based drug development. This work represents a significant step towards urgently needed therapeutic options for MERS-CoV, helping to mitigate the risks associated with potential future outbreaks and crises.

## Methods

### Ethics statement

Animal experiments were done by veterinarians with the authorization of animal operation and in accordance with the China law for animal protection. This study was approved by the Animal and Welfare Committee of Tianjin University (Approval number: TJUE-2024-053).

### Alpaca immunization

An alpaca (Female, 2–3 years old, Y- Clone, China, cat. AR-0019) was purchased from Xuzhou Animal Center (Xuzhou, China). 250 μg of the extracellular domain of MERS-CoV S protein fused with His tag (S1+S2 ECD) was emulsified with 250 μL Freund's complete adjuvant (F5881-10ML, Sigma) to immunize an alpaca. On weeks of 2, 6, and 9, the alpaca was boosted thrice with 250 μg S protein in 250 μL Freund's incomplete adjuvant (F5506-10ML, Sigma).

One week following the 4[th] immunization, 100 ml of blood was collected to measure anti-serum titer and construct a phage library displaying Nb.

## Cells lines

HEK-293T, HEK-293F, and Huh-7 cell lines used in the experiments were purchased from ATCC. The HEK-293T and Huh-7 cells were cultured in Dulbecco's modified Eagle's medium (DMEM) (Gibco) containing 10% fetal bovine serum (FBS) (Gibco), 100 U/ml penicillin, and 100 mg/ml streptomycin (Gibco) at 37˚C with 5% $CO_2$ in a humidified incubator. The HEK-293F were cultured at 37˚C, 5% $CO_2$, 220 rpm in a shaker, and in culture medium SMM 293-TI (cat. M293TI) containing 10% fetal bovine serum (FBS) (Gibco), 100 U/ml penicillin, and 100 mg/ml streptomycin (Gibco).

## Construction of a phage library displaying Nbs

Nb phage library was constructed following our previously published method with some modifications (Wu et al., 2020a). In brief, PBMCs were isolated from 100 ml blood of immunized alpaca using a lymphocyte separation solution (cat. 17-1140-02, Ficoll-Pa- que Plus, GE). RNA was extracted and reverse transcribed into cDNA by oligo (dT) and random hexamers as primers using the TRIzol kit (cat. 15596018, Life Technologies), following manufacturer's instruction. The alpaca Nbs gene were amplified with the combination of primers and cloned into phV1 phagemid plasmid (Y-Clone, Ltd., China) to transform TG1 bacteria.

## Panning Nb phage library and phage ELISA

The Nb-phagemid-transformed bacteria were rescued with M13KO7 helper phage (Invitrogen) and precipitated with PEG/NaCl. The phage Nb antibody library was enriched three times with 50 μg/ml of S protein. The enriched phage was eluted, transformed, and selected for the monoclonal phage to be evaluated by phage ELISA. 200 ng S protein in coating buffer (pH 9.6) was used to coat 96-well plates (Corning) at 4˚C overnight. After washing, the plates were blocked with blocking buffer (3% BSA in PBST; PBST, PBS with 0.02% Tween-20) for 1 h at 37˚C and then incubated with library phages or single clone phage in bacterial supernatant at 4˚C for 1.5 h. After washing, an anti-M13 bacteriophage antibody with HRP (1:10000 dilution, Sino Biological) was added and incubated at 37˚C for 1 h. Accordingly, TMB substrate (Sigma) was added at 37˚C for 10 min; 10 ml 0.2 M $H_2SO_4$ was added to stop the reaction. Optical densities were measured at 450 nm using the Infinite 200 (Tecan, Ramsey, MN, USA). Clones with readout at 450 nm > 5 were sequenced.

## Expression and purification of recombinant proteins

To purify and characterize the nanobodies, we cloned the Nbs gene into vector pcDNA3.4, and the Fc1 gene (CH2-CH3) of the human monoclonal antibody was fused on Nbs C-terminal. Transfected the different vectors into 293F cells to produce Nbs and harvested the cell culture after 72 h. Nb-Fcs were isolated and purified using Protein G (Thermo Scientific). We also expressed the monomer Nbs using the *E.coil* system. The Nbs gene was cloned into a pET22b vector containing a pelB signal peptide at the N-terminus and a hexa-His tag at the C-terminus, and the cloned vectors were transformed into BL21 competent cells for protein expression, and Nbs were isolated and purified using Ni-NTA (Thermo Fisher Scientific) with TBS buffer (25 mM Tris, 150 mM NaCl, pH 7.5).

The coding sequence of the MERS-CoV spike glycoprotein ectodomain (EMC strain, spike residues 1–1290) with a C-terminal hexa-His-tag was cloned into pcDNA3.1, and the S1

subunit (residues 18–751) with a C-terminal Fc-tag was cloned into the eukaryotic expression vector pVAX. FreeStyle 293-F cells were transfected with the plasmid using polyethyleneimine (PEI) (Sigma) and purified using Ni-NTA and Protein G, respectively.

The MERS-CoV RBD (residues 381–587) and NTD (residues 18–351) were expressed using the Bac-to-Bac baculovirus system. The constructs were transformed into bacterial DH10Bac component cells and extracted bacmids. We transfected the bacmids into sf9 cells using X-tremeGENE 9 DNA Transfection (Roche), harvested the low-titer viruses after 4 days, and then amplified to generate high-titer virus stock. Hi5 cells were infected with the high-titer virus and the cells culture containing the proteins were harvested after 72 h, concentrated and proteins were captured by Ni-NTA resin. The Ni-NTA resin was washed 4–5 times with wash buffer (25 mM Tris, 150 mM NaCl, 30 mM imidazole, pH 7.5), and the target proteins were eluted with elution buffer (25 mM Tris, 150 mM NaCl, 300 mM imidazole, pH 7.5). The target proteins were then purified on Superdex 75 High-Performance column (GE Healthcare) with TBS (25 mM Tris, 150 mM NaCl, pH 7.5). SDS-PAGE analysis revealed the target proteins were over 95% purity, and then concentrated the target proteins to 15 mg/ml for subsequent experiments.

## ELISA and Ni-NTA pull-down assays for the specific recognized region of Nbs

200 ng of S1, NTD, and RBD proteins were used to coat 96-well plates at 4˚C overnight. The plates were washed and blocked with blocking buffer (3% BSA in PBST) for 1 h at 37˚C, then incubated with Nb-Fcs proteins at 4˚C for 2 h. After washing, an anti-Fc antibody conjugated with HRP (1:10000 dilution, Sino Biological) was added and incubated at 37˚C for 1 h, and then the TMB substrate (Sigma) was added at 37˚C for 10 min. 10 ml of 0.2 M $H_2SO_4$ was added to stop the reaction. Optical densities were measured at 450 nm using the Infinite 200 (Tecan, Ramsey, MN, USA), and the results were analyzed using GraphPad Prism 9 (GraphPad Software Inc.).

The MERS-CoV NTD and RBD were expressed using Bac-to-Bac baculovirus system, and the NTD and RBD with C-terminal 6×His tag. Hi5 cells were infected with the high-titer virus and harvested the 2 ml of cell culture in six-well plates after 72 h. The proteins were captured by Ni-NTA resin and buffer exchanged to PBS. Then, incubated with untagged nanobodies for 4 h, and washed by PBS containing 20 mM imidazole. The bound proteins were eluted with PBS containing 300 mM imidazole, and the eluted sample was mixed with loading buffer boiled at 100˚C for 5 min, and then loaded on 15% SDS-PAGE gel. The gel was stained with coomassie blue.

## Complex preparation and crystallization

The MERS-CoV RBD and the Nbs (Nb14 and Nb9) were mixed at a molar ratio of 1:1.5, incubated at 4˚C for 3 h, and then purified by Superdex 75 (GE Healthcare). 10 mg/ml and 20 mg/ml of MERS-CoV RBD/Nbs were performed for crystal screening by mixing 0.5 μl of protein with 0.5 μl of reservoir solution at 16˚C using the vapor-diffusion sitting-drop method. The complex of Nb14-MERS-CoV RBD rode-like crystals appeared after four days at the PEGRx Kits and Crystal Screen Kits (Hampton), and the final optimized diffraction crystals at mother liquid containing 10% w/v PEG 6000, 8% Ethylene glycol, 0.1 M Citric acid pH 3.5 by the hanging-drop vapor-diffusion method. The Nb9-MERS-CoV RBD crystal was successfully grown in WIZARD IV (Emerald BioSystems). Further optimization with additive and hanging-drop vapor-diffusion method was performed. The final crystals used for diffraction data

collection were obtained in 24% w/v PEG 1500, 100mM SPG buffer pH 8.4, and were dehydrated and cryo-protected in 4 M Sodium formate.

## Data collection, processing, and structure determination

Crystals were flash-cooled in liquid nitrogen after being incubated in reservoir solution containing 20% (v/v) glycerol, and diffraction data were collected at the Shanghai Synchrotron Radiation Facility (SSRF). The Nb14-MERS-CoV RBD complex crystals were collected at BL02U1 (wavelength, 0.97918 Å) at 100K, and the Nb9-MERS-CoV RBD complex crystals were collected at BL10U2 (wavelength, 0.97856 Å) at 100K. All data were processed using the HKL3000 package [53], and structures were solved by molecular replacement using PHASER with the MERS-CoV RBD structure (PDB ID: 4L3N) and the structure of the Nanobody we previously defined (PDB ID: 7X7E). The subsequent models were built and refined by COOT and PHENIX, respectively [54–56]. Model geometry was verified using the MolProbity program, and the structure refinement statistics are listed in S2 Table. The epitope and critical residues were identified by PISA and LigPlot$^+$ 2.2 at the European Bioinformatics Institute, and all structure figures were generated by PyMOL (http://www.pymol.org).

## Neutralization activity of Nbs against MERS-CoV pseudovirus

Pseudoviruses carrying the full-length spike envelope of prototype MERS-CoV were generated as previously reported [25,57]. Specifically, human immunodeficiency virus (HIV) backbones expressing firefly luciferase (pNL4-3-R-E-luciferase) and pcDNA3.1 vector encoding either MERS-CoV S proteins or mutations were co-transfected into the HEK-293T cells (ATCC). Pseudoviruses in the viral supernatants were collected after 48 h, centrifuged to remove cell lysis, measured the viral titers by luciferase assay in relative light units (Bright-Glo Luciferase Assay Vector System, Promega Biosciences), and stored at -80°C until used. The wildtype pseudovirus used throughout the analysis was the prototype strain (GeneBank: APB87331), and the mutagenesis based on wildtype genes containing 8 changing residues: L411F, T424I, L506F, D509G, I529T, V530L, E536K, D537E (Fig 5A). For neutralization assay, the Nb22 against SARS-CoV-2 served as a control [46]. The 50% tissue culture infectious dose (TCID$_{50}$) was determined by infection of Huh-7 cells. Neutralization activity assays of Nbs were performed by incubating pseudoviruses with serial dilutions of purified Nbs or serums at 37°C for 1 h, and then the virus-nanobodies mixtures were added to the Huh-7 cells. After incubation for 72 h at 37°C, the neutralizing activities of nanobodies were determined by the luciferase activity and presented as half-maximal inhibitory concentrations (IC$_{50}$), calculated using the dose-response inhibition function in GraphPad Prism 9 (Graph-Pad Software Inc.).

## FACS analysis of cell-surface staining

Nbs binding affinities were analyzed with complex cell surface by fluorescence-activated cell sorting (FACS). The cell-surface staining experiments were performed at room temperature and tested by Flow cytometry (BD eBiosciences). HEK-293T cells, which overexpressed the S protein, were stained by Nb-Fcs and incubated for 30 min. Cells were washed three times with PBS (Solarbio), and then stained by anti-Fc-APC (Abcam) for 30 min. Cells were subsequently washed with PBS five times and analyzed by flow cytometry. The binding between recombinant soluble MERS-CoV RBD and hDPP4 expressed on the surface of Huh-7 cells was also measured using FACS. The MERS-CoV RBD with His-tag was incubated with Nbs in advance at molar ratios 1:1 for 30 min and then incubated the mixtures with Huh-7 cells for 30 min. After washing the unbound RBD with PBS three times, the Huh-7 cells were then stained with

anti-His-APC (Abcam) for another 30 min. After staining, the cells were washed with PBS five times and then analyzed by flow cytometry on a FACS Aria III machine (BD eBiosciences).

## Affinity determination by Bio-Layer Interferometry (BLI)

Affinity assays were performed on an Octet R8 biolayer interferometry instrument (Sartorius, Germany) at 25°C with shaking at 1,000 rpm. To measure the affinity of Nbs with human Fc tag, anti-human Fc (AHC) biosensors (cat.# 18–5060, Sartorius) were hydrated in water for 30 min prior to 60 s (sec) incubation in a kinetic buffer (PBS, 0.02% (v/v) Tween-20, pH 7.0). Either Nb-Fc in cell supernatant or purified Nbs-Fc were loaded in a kinetic buffer for 200 s prior to baseline equilibration for 200 s in a kinetic buffer. The data were baseline subtracted before fitting was performed using a 1:1 binding model and the Octet R8 data analysis software. $K_D$, $K_a$, and $K_d$ values were evaluated with a global fit applied to all data.

## Cooperativity of the two nanobodies for virus neutralization

The FACS analyzes the cooperativity between two nanobodies for blocking RBD binding with human hepatoma Huh-7 cells. The MERS-CoV RBD with His-tag was incubated with Nb9 in advance at molar ratios 1:1 at 37°C for 30 min and divided the mixture into two parts. Nb14 was added into one of the mixtures and incubated at 37°C for 30 min. Huh-7 cells were stained by the different mixtures at 37°C for 30 min, washed with PBS three times, and then stained with anti-His-APC for 30 min. Cells were washed five times and then analyzed by flow cytometry. The data of the different experimental groups were statistically analyzed and processed by GraphPad Prism 9.

The synergistic, additive, and antagonistic interactions between Nb14 and Nb9 in virus neutralization were assessed using the median effect analysis method implemented in the CompuSyn software [26]. The determined neutralization values were incorporated into the program as fractional effects (FA), ranging from 0.01 to 0.99 for each Nb individually and for their combined application. The computation of Combination Index (CI) values were conducted relative to the corresponding FA values. A logarithmic CI value of 0 denotes an additive effect, <0 indicates synergism, and >0 suggests antagonism.

## Supporting information

**S1 Fig. Characterization of anti-sera specific for MERS-CoV S protein.** (**A**). The experimental schedule of immunization. (**B**). The titer of anti-sera was evaluated by ELSIA after the 4th immunization in alpaca receiving MERS-CoV S protein. Y-axis represented the absorbance at 450 nm, X-axis was the anti-sera dilution fold. 4th anti-sera represented alpaca anti-sera after the 4th immunization of MERS-S protein (cyan line). Pre-Immun. was the sera from the alpaca before the immunization of MERS-CoV S protein. The pound sign indicated the anti-sera titer. (**C**). Anti-sera inhibited MERS-CoV from infecting Huh-7 cells. The line color was indicated as (**B**).
(TIF)

**S2 Fig. The biopanning and screening of MERS-VHH phage library.** (**A**). The summary of MERS–VHH-library. (**B**). The binding of phage library with MERS-CoV S protein identified by phage ELISA, Library is the phage library of MERS-VHH. 1st, 2nd, and 3rd are the phage library after panning on one round, two rounds, and three rounds of MERS-CoV S protein enrichment, respectively. (**C**). Positive clones of phage from MERS-VHH library after the 1st, 2nd, and 3rd enrichment of MERS-S protein. One dot represented a single clone. (**D**). The summary of bacterial supernatant binding with MERS- CoV S protein tested by phage ELISA. The

Y-axis was the ratio of the readout of MERS-CoV S binding/the readout of blank binding. One dot represented bacterial supernatant from single clone. The ratio above 5 was taken as positive binders. Among 182 clones, there were 49 positive binders.
(TIF)

**S3 Fig. Characterizing the binding of Nbs. (A)**. The affinity summary of Nbs binding with MERS-S1 protein tested by BLI. $K_{on}$, association rate constant, $k_d$, dissociate constant; $K_D$, binding affinity; $R_{Max}$, the max response unit. (**B**). Nb-Fcs binding with MERS-CoV S which overexpressed in HEK-293T cells and analyzed by FACS, associated with Fig 2A. (**C**). Epitope analysis of Nbs by BLI. MERS-S1 protein was coated on the sensor, and Nb9 as the first antibody was added to bind for 200 s, followed by Nb11, Nb14, and Nb67 as the second antibody for another 200 s.
(TIF)

**S4 Fig. The complexes of Nbs and RBD were purified by Superdex 75.** The gel chromatography and SDS-PAGE analyses of RBD and Nb9 (**A**), Nb11 (**B**), Nb14 (**C**) and Nb67 (**D**).
(TIF)

**S5 Fig. The recognition motif analyses of Nb14 and Nb9.** (**A**). The sequence alignment of Nb9 and Nb14. The red triangles represented the residues of Nb14 interacted with RBD, and the blue dots represented the residues of Nb9 interacted with RBD. (**B**) and (**C**). The residues of RBD interacted with Nb9 (blue dots) and Nb14 (red triangles). (**D**). The CCL loop of Nb14 interacted with the β4-β5 loop of RBD. (**E**). the hydrophobic network of Nb14 CCL loop and RBD β4-β5 loop. (**F**). The residues R59 and Y37 of Nb9 FRs formed hydrogen bonds with Q466 and E513 of RBD.
(TIF)

**S6 Fig. The neutralizing activities of Nb14 and Nb9 against pseudotyped MERS-CoV with natural mutations.** The $IC_{50}$ of Nb14 (**A**) and Nb9 (**B**) against different pseudotyped MERS-CoV with natural mutations.
(TIF)

**S7 Fig. The influence of mutations on Nb14 and Nb9 paratopes for Nbs and RBD bound.** (**A**). The key residues of Nb14 paratope. (**B**). The key residues of Nb9 paratope. All the CDRs are shown as yellow (CDR1), violet (CDR2), and red (CDR3). (**C**). The binding affinities of Nb14 mutations for RBD bound. (**D**). The binding affinities of Nb14 mutations for RBD bound. Statistics are summarized in S5 Table.
(TIF)

**S8 Fig. Sequence alignment of MERS-CoV RBD natural mutations.**
(TIF)

**S9 Fig. The reported monoclonal antibodies targeted at MERS-CoV RBD were divided into three categories.** (**A**). The epitopes of three classes are circled by green (Class I), black (Class II), and red (Class III) rings. (**B**). The epitope of Nb14 is out of the three Classes. The epitope of Nb9 is partially overlapped with the Class III epitope. (**C-E**). Structural superimpose between Class I, Class II, and Class III antibodies with hDPP4-RBD. (**F-H**). Structural superimposed between Class I, Class II, and Class III antibodies with MERS-CoV S which with all RBD in the "down" positions. CDC2-C2 (PDB: 6C6Z); m336 (PDB: 4XAK); 4C2 (PDB: 5DO2); D12 (4ZPT); MERS-4V2 (PDB: 5YY5); hDPP4: (4L72); MERS-CoV S (PDB: 5w9j).
(TIF)

**S10 Fig. The FACS results histogram of Fig 6B.** (**A**). Nb14 and Nb9 inhibit the binding of RBD and Huh-7 cells, and the FACS peak shifts forward. RBD: Red; RBD+Nb14: green; and RBD+Nb9: cyan. (**B**). The histogram results of Nb14 and Nb9 inhibition. Data are represented as mean ± SD from three independent experiments.
(TIF)

**S11 Fig. The Synergistic neutralizing effects of Nb14-hFc1 and Nb9-hFc1.** (**A**). The SDS-PAGE analyses of Nb9-hFc1 and Nb14-hFc1. (**B**). Neutralizing effects of Nb14-hFc1 combined with Nb9-hFc1 against pseudotyped MERS-CoV.
(TIF)

**S1 Table. Data collection and refinement statistics.**
(XLSX)

**S2 Table. Residues contributed to the interaction between Nb14-RBD.**
(XLSX)

**S3 Table. Hydrogen bonds of interaction between Nb9-RBD.**
(XLSX)

**S4 Table. Neutralizing activities of Nbs against infection of MERS-CoV pesudoviruses.**
(XLSX)

**S5 Table. The affinity of the binding between Nbs-mutation and RBD.**
(XLSX)

**S1 Data. Source data used to generate all graphs in this article.**
(XLSX)

## Acknowledgments

We acknowledge Y-Clone Company for their support in analyzing the binding or epitope binning of Nbs, and SSRF BL02U1 and BL10U2 beam lines for crystal data collection.

## Author Contributions

**Conceptualization:** Sen Ma, Yaxin Wang.

**Data curation:** Sen Ma, Doudou Zhang, Linjing Zhu.

**Formal analysis:** Sen Ma, Qiwei Wang.

**Funding acquisition:** Sheng Ye, Yaxin Wang.

**Methodology:** Sen Ma, Yaxin Wang.

**Project administration:** Sheng Ye, Yaxin Wang.

**Resources:** Sen Ma, Xilin Wu, Yaxin Wang.

**Software:** Sen Ma.

**Supervision:** Xilin Wu, Sheng Ye, Yaxin Wang.

**Validation:** Sen Ma, Doudou Zhang, Linjing Zhu.

**Visualization:** Yaxin Wang.

**Writing – original draft:** Sen Ma, Yaxin Wang.

**Writing – review & editing:** Sen Ma, Sheng Ye, Yaxin Wang.

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
