## [Decision Letter · Decision Letter 0]

1 Jul 2024

Dear Dr. Wang,

Thank you very much for submitting your manuscript "Structure defining of ultrapotent neutralizing nanobodies against MERS-CoV with novel epitopes on receptor binding domain" for consideration at PLOS Pathogens. As with all papers reviewed by the journal, your manuscript was reviewed by members of the editorial board and by several independent reviewers. The reviewers appreciated the attention to an important topic. Based on the reviews, we are likely to accept this manuscript for publication, providing that you modify the manuscript according to the review recommendations.

The reviewers were very positive about your manuscript and feel the study of these antibodies will be of interest. There are however some issues they would like you to address, most are minor. Please respond to each of the reviewers comments in your resubmission.

Sincerely,

Julie Overbaugh

Guest Editor

PLOS Pathogens

Guangxiang Luo

Section Editor

PLOS Pathogens

Michael Malim

Editor-in-Chief

PLOS Pathogens

orcid.org/0000-0002-7699-2064

The reviewers were very positive about your manuscript and feel the study of these antibodies will be of interest. There are however some issues they would like you to address, most are minor. Please respond to each of the reviewers comments in your resubmission.

Reviewer Comments (if any, and for reference):

Reviewer's Responses to Questions

**Part I - Summary**

Reviewer #1: Ma, et al, studied four potent nanobodies against MERS-CoV isolated from alpaca. Therein, Nb14 and Nb9 targets the different cryptic face of RBD, and that Nb14 framework regions are mainly involved in interactions that target a novel epitope, which is entirely distinct from all three class antibodies. The discovery of these Nbs holds great promise in clinical therapeutics, offering innovative solutions in response to the persistent and evolving challenges posed by coronaviruses. It is a topic of interest to the researchers in the related areas but the paper needs very some improvement before acceptance for publication.

Reviewer #2: In this manuscript, Ma et al, utilized the recombinant MERS-CoV S protein to immunize the alpaca and successfully isolated four nanobodies with ultrapotent neutralizing activity. The authors further defined the crystal structures of Nb14-/Nb9-RBD and elucidated the molecular mechanism of how Nb14 and Nb9 work. Notably, among these four nanobodies, Nb14 exhibited the most prominent activity and specifically targeted a novel epitope. This study provides a highly valuable research basis for the combination of two nanobodies against MERS-CoV, leveraging the distinct recognition epitopes and mechanisms of action exhibited by Nb14 and Nb9, and provides insights for the development of broad-spectrum nanobodies against MERS-CoV. Overall, this work is very interesting and important. This reviewer believe this manuscript merits publication, if the authors can address the following concerns.

Reviewer #3: Ma et al. describe the identification and characterization of four nanobodies against MERS-CoV. Two of these are studied in detail using a range of techniques to demonstrate unique binding and neutralizing capacities. The mechanisms of neutralization identified, either blocking the RBD in the upward postion or RBD-glycan interactions add to the understanding of the interaction of MERS-CoV spike protein with DPP4. Mutants made, both RBD as well as DPP4, add to the further characterization next to the structural studies. Overall these are intersting findings.

**Part II – Major Issues: Key Experiments Required for Acceptance**

Reviewer #1: (No Response)

Reviewer #2: 1.The authors evaluated the synergistic effect of the combination of Nb14 and Nb9 based on the different epitopes. Nanobody traditionally requires fusing human Fc to extend the half-life in the human body. Therefore, will Nb14 and Nb9 retain the same synergistic effect when fused with Fc? The authors should test, or at least discuss this issue in their revision.

2. In Figure 6B, the FACS results should be consistent with that of Figure 8B, and the data of different experimental groups should be counted as a histogram.

Reviewer #3: My major concern is the gramma used. The article should be fully checked. In all sections, including the tittle, abstract as well as the results section many corrections should be made.

**Part III – Minor Issues: Editorial and Data Presentation Modifications**

Reviewer #1: 1. P9: “The library had a size of 1.37 × 10^9, exhibiting a remarkable sequence diversity of 100% with an impressive in-frame rate of 84%, which was rigorously validated through PCR and sequencing.”

10^9 should be 109 for consistency.

2. P10 and P11: “The neutralizing activity of Nbs was dosedependent in Huh-7 cells, and with stronger potency of Nb14 (IC50, 0.0014 μg/ml), Nb11 (IC50, 0.0022 μg/ml), and Nb67 (IC50, 0.0025 μg/ml), respectively (Fig. 2E and 2H). In contrast, Nb9 displayed a slightly weaker neutralizing activity with an IC50 of 0.016 µg/ml (Fig. 2E and 2H).”

According to Fig. 2H, the IC50 of Nb11 should be 0.0023 μg/ml and the IC50 of Nb9 should be 0.0165 μg/ml.

3. P19: “The FACS showed that Nb9 and Nb9+Nb14 inhibition rates, which blocked the staining of Huh-7 cells by the RBD, were 80% and 97%, respectively (Fig. 8B).”

According to Fig. 8B, the inhibition rates of Nb9+Nb14 is about 93%?

How about the inhibition rates of Nb14 alone? According to Fig. 6B, the inhibition rates of Nb14 is about 93% and is close to the inhibition rates of Nb9+Nb14. In this way, dose these results fully illustrate that Nb14 could cooperate with Nb9 due to differences in epitopes?

Reviewer #2: 1. In Figure 5, the authors evaluated the neutralizing activity of Nb14 and Nb9 against MERS-CoV strains with several natural mutations. The authors should add a sequence alignment diagram to show the location of each mutation on RBD of MERS-CoV strains.

2. “We similarly compared the binding of Class I, Class II, and Class III antibodies to the closed state of MERS-CoV S by structure superimpose.” In this part, the “MERS-CoV S” means the S trimer?

3. “Although we could not character the atom level” should be “atomic level”.

4. “Statistics on diffraction data collection” should be “Statistics of”.

5. “CDR2 and CDR3 of Nb14 binding with RBD” should be “are binding”.

6. “Particularly, these atural changes, named L506, D509, and E536”, the “atural” should be “natural”.

7. “Binding the hDPP4 host receptor and activating the S trimer is necessary for virus infection”, the “S trimer” should be “S trimers”.

8. “Our findings indicate that while most antibodies exhibited limited effectiveness in binding to the closed-state MERS-CoV S”. In this section, the “MERS-CoV S” expression should be “MERS-CoV S trimers”, consistent with the previous.

9. “The MERS-CoV NTD and RBD were express Bac-to-Bac baculovirus system” should be “were expressed”.

10. “Nb14 were added into one of the mixtures and incubated at 37 ℃ for 30 min” should be “Nb14 was added”.

11. Figure 2 legends “(E, F). Nb9, Nb11, Nb14, and Nb67could effectively neutralize MERS-CoV pseudovirus in vitro” the “Nb67could” should be “Nb67 could”.

Reviewer #3: A few minor issues need to be addressed:

1. Intro: MERS-CoV did not cause a pandemic...

2. SARS-CoV-1 should be replaced by SARS-CoV

3. When analyzing different mutations in the RBD in pseudotypes small drops in neutralizing capacity are observed. However, variation in the IC50 values against WT virus are not shown.

4. page nrs or lines are not provided which makes it difficult to provide accurate feedback...

PLOS authors have the option to publish the peer review history of their article (what does this mean?). If published, this will include your full peer review and any attached files.

Reviewer #1: No

Reviewer #2: No

Reviewer #3: No

Figure Files:

Data Requirements:

Reproducibility:

References:

---

## [Editor Report · Decision Letter 1]

22 Jul 2024

Dear Dr. Wang,

We are pleased to inform you that your manuscript 'Structure defining of ultrapotent neutralizing nanobodies against MERS-CoV with novel epitopes on receptor binding domain' has been provisionally accepted for publication in PLOS Pathogens.

Best regards,

Julie Overbaugh

Guest Editor

PLOS Pathogens

Guangxiang Luo

Section Editor

PLOS Pathogens

Michael Malim

Editor-in-Chief

PLOS Pathogens

orcid.org/0000-0002-7699-2064

The authors have addressed the reviewers' concerns
---

## [Editor Report · Acceptance letter]

30 Jul 2024

Dear Dr. Wang,

We are delighted to inform you that your manuscript, "Structure defining of ultrapotent neutralizing nanobodies against MERS-CoV with novel epitopes on receptor binding domain," has been formally accepted for publication in PLOS Pathogens.

Best regards,

Michael Malim

Editor-in-Chief

PLOS Pathogens

orcid.org/0000-0002-7699-2064